# Sesquiterpenoids and Their Anti-Inflammatory Activity: Evaluation of *Ainsliaea yunnanensis*

**DOI:** 10.3390/molecules24091701

**Published:** 2019-05-01

**Authors:** Jinjie Li, Xiuting Li, Xin Wang, Xiangjian Zhong, Linlin Ji, Zihan Guo, Yiran Liu, Xiaoya Shang

**Affiliations:** 1Beijing Key Laboratory of Bioactive Substances and Functional Foods, Beijing Union University, Beijing 100191, China; lijinjie.7785004@163.com (J.L.); shtwangxin@buu.edu.cn (X.W.); xiangjzhong@163.com (X.Z.); jll5927@163.com (L.J.); 15910660409@163.com (Z.G.); hunter88rising@163.com (Y.L.); 2Beijing Advanced Innovation Center for Food Nutrition and Human Health, Beijing Technology and Business University, Beijing 100048, China; lixt@btbu.edu.cn

**Keywords:** *Ainsliaea yunnanensis*, sesquiterpenoids, structures, inflammasome

## Abstract

Four new sesquiterpenoids (**1**–**4**) and six known sesquiterpenoids (**5**–**10**), were isolated from the EtOAc phase of the ethanolic extract of *Ainsliaea yunnanensis*. Their structures were established by spectroscopic methods, including 1-D, 2-D NMR and HPLC-MS. All compounds were tested for their anti-inflammatory effect by the inhibition of the activity of NLRP3 inflammasome by blocking the self-slicing of pro-caspase-1, which is induced by nigericin, then the secretion of mature IL-1β, mediated by caspase-1, was suppressed. Unfortunately none of the compounds showed an anti-inflammatory effect.

## 1. Introduction

The genus *Ainsliaea* belongs to the family Compositae and comprises around 70 species distributed primarily in southeastern Asia [1]. More than 20 species are used in medicine to treat colds, coughs and asthma, rheumatism, numbness and pain, injury, blood circulatory disorders and hemostasis, enteritis and dysentery, laryngitis, and urinary and gynecological diseases [2]. Modern pharmacological studies have revealed varied biological activities, such as anti-inflammatory [3] and antitumor effects [4]. *Ainsliaea yunnanensis* is a widely used species in this genus, and is known as “zhui feng jian”, ”yan mai ling” and ”bone arrow” in folk, and is also the common folk herb used for treating various disorders [5].

Previous phytochemical studies of this species have led to the isolation and identification of sesquiterpenoids, triterpenoids, isovaleryl sucrose esters and three other components [6,7,8,9,10,11]. Sesquiterpenoids are an important type of constituent contributing to bioactivity, and have been revealed to possess potential anti-inflammatory activity [6,8]. During our ongoing search for bioactive natural products, we found that the EtOAc phase of *Ainsliaea yunnanensis* showed potential anti-inflammatory activity (see Appendix A). In order to investigate the additional minor sesquiterpenoids with novel structures and anti-inflammatory activity from *Ainsliaea yunnanensis*, bioassay-guided fractionation led to the isolation of four new minor sesquiterpenoids: ainsliatone A acid (**1**), alatoside M (**2**), alatoside N (**3**), 4β,15-dihydrozaluzanin C (**4**), and six known abundant sesquiterpenoids: ainsliatone A (**5**), 8α-hydroxy-11α,13-hydroxyzaluanin C (**6**), 3,8-hydroxy-10(14)-guaien-6,12-olide (**7**), ixerin W (**8**), glucozaluzanin C (**9**), and picriside B (**10**). The structures of the sesquiterpenoids were elucidated by spectroscopic methods and compared with those reported in the literature (Figure 1).

Abnormal activation of the inflammasome complex can lead to a variety of immune inflammatory diseases. In recent years, MCC950 exerted its anti-inflammatory effect by inhibiting the activation of the NLRP3 inflammasome [12]. Therefore, the development of small molecular compounds that can regulate the inflammasome will be of great significance for the prevention and treatment of inflammatory diseases. As a consequence, activation assays of the NLRP3 inflammasome of these sesquiterpenoids have been carried out.

## 2. Results

The EtOH extract of *Ainsliaea yunnanensis* was partitioned between water and petroleum ether, EtOAc, and normal butanol. The EtOAc phase was concentrated under a vacuum (<50 °C) and then separated repeatedly by the various chromatographic separation and purification methods to obtain compounds **1**–**10** (Figure 1, the detailed data are in the Appendix A).

### 2.1. Structural Elucidation of New Compounds

Compound **1** was obtained as a white amorphous powder. The HR-ESIMS at *m*/*z* 269.1382 [M + H]^+^ (calcd. 269.1382) indicated the molecular formula of **1** as C_14_H_20_O_5_. The ^1^H-NMR spectrum of **1** in CD_3_OD displayed signals for one singlet methyl group (δ_H_ 0.76), two oxygenated methines (δ_H_ 3.79 (1H, dd, *J* = 5.0, 10.0 Hz); 4.14 (1H, t, *J* = 10.0 Hz)) and two exocyclic olefinic signals (δ_H_ 5.67, 6.23). The ^13^C-NMR spectrum showed 14 carbon signals, which were classified as one methyl, five methylenes (one olefinic methylene), four methines (two oxygenated methylenes), and four nonprotonated carbons (one carbonyl, one carboxyl and one olefinic carbon) based on HSQC spectra (Table 1). All spectroscopic data above in combination with the five degrees of unsaturation required by the molecular formula suggested that **1** was a dual ring containing one double bond, one carbonyl group and one carboxyl group. The ^1^H-NMR and ^13^C-NMR signals in **1** were very similar to those of the known compound ainsliatone A (**5**) [13]. However, the molecular formula established by using HR-ESIMS in compound **1** (C_14_H_20_O_5_) and **5** (C_14_H_18_O_4_) are very different. The structure of **1** was finally characterized by the careful analysis of its 2D-NMR spectroscopic data including ^1^H-^1^H COSY and HMBC (Figure 2).

Two structural fragments, as shown with bold lines in Figure 2 (C-1 through C-2 to C-3 and C-5 through C6 up to C-9), were first established by the correlations observed in the ^1^H-^1^H COSY spectrum. The connectivity of the two structural fragments, one dual ring and the other functional groups were mainly achieved by the analysis of the HMBC spectrum (Figure 2). A very long-range HMBC correlation from H-5 to C-1, C-3, C-4, C-6, C-7, C-9, and C-10, along with their chemical shifts, not only indicated the structure of the dual ring but also the position of the two hydroxyl groups. The other long correlation from CH_3_-14 to C-1, C-5, C-9 and C-10 located the CH_3_-14 at C-10. In addition, an important correlation from H-7 to C-11, C-12, and C-13 confirmed the structure of the side chain. The planar structure of **1** was, therefore, determined as ainsliatone A acid (Figure 1).

The relative stereochemistry of **1** was elucidated by the analysis of its NOESY data and compared with that of the known compound ainsliatone A [13] as shown in Figure 2. NOE correlations between H-1 and H-5, and H-5 and H-7 were observed. Thus H-1, H-5, and H-7 adopted the same orientation and were arbitrarily designated as the α-orientation. The large coupling constant between H-5 and H-6 (J_H-6/H-5_ = 10.0 Hz) suggested a trans configuration and therefore H-6 should be in the β-orientation. The correlation between H-6 and CH_3_-14 suggested the β-orientation of the methyl at C-14. Accordingly, the structure of 1 was established as ainsliatone A acid.

Compound **2** was obtained as a white amorphous powder. The HR-ESIMS at m/z 435.1985 [M + Na]^+^ (calcd. 435.1989) indicated the molecular formula of **2** as C_21_H_32_O_8_. The ^1^H-NMR spectrum of **1** in CD_3_OD displayed the signals for two singlet methyl groups (δ_H_ 0.85, 1.58), one cyclic olefinic signal (δ_H_ 5.31), two exocyclic olefinic signals (δ_H_ 5.58, 6.14) and a signal attributable to the anomeric proton of the glucosyl moiety (δ_H_ 4.30). The ^13^C-NMR spectrum showed 21 carbon signals, which were classified as four olefinic carbons (δ_C_ 120.6, 123.0, 136.2, 148.0), one carboxyl group (δ_C_ 170.9), two methyl groups (δ_C_ 10.9, 21.0), one oxygenated methylenes (δ_C_ 82.2) and one glucose moiety (δ_C_ 101.5, 78.2, 77.8, 75.2, 71.9, 63.0) based on HSQC spectra (Table 1). All spectroscopic data above in combination with six degrees of unsaturation required by the molecular formula suggested that **2** was a dual ring containing two double bonds, one carboxyl group and one glucose. The ^1^H-NMR and ^13^C-NMR signals in **2** were very similar to eudesmane-type sesquiterpene glycoside [14]. The structure of **2** was finally characterized by the careful analysis of its 2D-NMR spectroscopic data including ^1^H-^1^H COSY and HMBC (Figure 1).

Two structural fragments as shown with bold lines in Figure 3 (C-2 through C3 to C-4 and C-6 through C7 up to C-10) were first established by the correlations observed in the ^1^H-^1^H COSY spectrum. The connectivity of the two structural fragments, one dual ring and the other functional groups were mainly achieved by the analysis of the HMBC spectrum (Figure 3). Long-range HMBC correlations from CH_3_-15 to C-4, C-5, C-6 and C-10, CH_3_-14 to C-1, C-2 and C-10 combining their chemical shifts not only indicated the structure of the dual ring but also located the CH_3_-14 at C-1, CH_3_-15 at C-5, respectively. In addition, two important correlations from H-4 to Glu-1 and C-6, confirmed the position of the glucose, and from H-7 to C-11, C-12, and C-13 confirmed the structure of the side chain. The planar structure of **2** was, therefore, determined as 4-*O*-(d-glucopyranosyloxyl) -eudesma-1,11(13)-dien-12-oic acid (Figure 1).

The relative stereochemistry of **2** was elucidated by the analysis of its NOESY data and compared with the known literature [14] as shown in Figure 3. NOE correlations between H-4 and H-10, and H-10 and H-7 were observed. Thus H-4, H-7, and H-10 adopted the same orientation and were arbitrarily designated as the α-orientation. Acid hydrolysis of **2** afforded D-glucose which was determined by GC-MS of methanolysate and silylated derivatives, and the characteristic coupling constant of the anomeric proton (*J* = 8.0 Hz) indicated that it was a β-d-glucoside. Accordingly, the structure of **2** was established as 4β-*O*-(β-d-glucopyranosyloxyl)-eudesma-1,11(13)-dien-12-oic acid, and named alatoside M.

Compound **3** was obtained as white amorphous powder. The HR-ESIMS at m/z 379.2090 [M + Na]^+^ (calcd. 379.2091) indicated the molecular formula of **3** as C_19_H_32_O_6_. The ^1^H-NMR and ^13^C-NMR signals in **3** were very similar to those of compound **2** except that **3** had one additional double methyl group (δ_H_ 1.13) and a lack of two exocyclic olefinic signals (δ_H_ 5.58, 6.14), two olefinic carbons (123.0, 148.0) and one carboxyl group (170.9) (Table 1). Acid hydrolysis of **3** afforded d-glucose, which was determined by GC-MS of methanolysate and silylated derivatives, and the characteristic coupling constant of the anomeric proton (*J* = 7.5 Hz) indicated that it was a β-d-glucoside. So the structure of **3** was elucidated as 4β-*O*-(β-d-glucopyranosyloxyl)-eudesma-7-methyl-1-olefin, and named alatoside N (Figure 1).

Compound **4** was obtained as a white amorphous powder. The HR-ESIMS at m/z 433.1832 [M + Na]^+^ (calcd. 433.1833) indicated the molecular formula of **4** as C_21_H_30_O_8_. The ^1^H-NMR spectrum of **1** in DMSO displayed signals for one singlet methyl groups (δ_H_ 1.15), four exocyclic olefinic signals (δ_H_ 6.00, 5.59, 4.99, 4.95), two oxygenated methines (δ_H_ 3.56, 3.93) and a signal attributable to the anomeric proton of the glucosyl moiety (δ_H_ 4.19). The ^13^C-NMR spectrum showed 21 carbon signals, which were classified as four olefinic carbons (δ_C_ 112.7, 119.5, 140.1, 149.5), one carboxyl group (δ_C_ 169.6), one methyl group (δ_C_ 18.1), two oxygenated methylenes (δ_C_ 86.1, 86.2) and one glucose moiety (δ_C_ 104.1, 76.8, 76.8, 73.6, 70.2, 61.2) based on HSQC spectra (Table 1). All spectroscopic data above in combination with seven degrees of unsaturation required by the molecular formula suggested that **4** was three rings containing two double bonds, one carboxyl group and one glucose. The ^1^H-NMR and ^13^C-NMR signals in **4** were very similar to compound **9**, belonging to guaiane-type sesquiterpene glycoside [15]. However, the quantity of the olefinic carbons in **4** (four olefinic carbons) and **9** (six olefinic carbons) is very different. In the ^1^H-NMR spectrum of **4**, the signals of the exocyclic methylene protons at C-15 were replaced by methyl. An important correlation from CH_3_-15 to C-3, C-4 and C-5 confirmed the position of the methyl. Acid hydrolysis of **4** afforded d-glucose, which was determined by GC-MS of methanolysate and silylated derivatives, and the characteristic coupling constant of the anomeric proton (*J* = 7.5 Hz) indicated that it was a β-d-glucoside. The structure of **4** was finally characterized by the careful comparison to spectroscopic data of compound **4** and **9** including ^1^H-^1^H COSY and HMBC (Figure 1), which was named 3-*O*-β-d-glucopyranosyloxyl -1α, 5α, 7α-H-10(14), 11(13)-dien-12, 6α-guaiactone (4β, 15-dihydrozaluzaninC) (Figure 1).

### 2.2. Anti-Inflammatory Activity Assay of Compounds

Compounds **1**–**10** were tested for their anti-inflammatory effect through the inhibition of the activity of the NLRP3 inflammasome by blocking the self-slicing of pro-caspase-1, which was induced by nigericin, then the secretion of mature IL-1β, mediated by caspase-1, was suppressed. Unfortunately none of the compounds showed any anti-inflammatory effect (Figure 4).

## 3. Discussion

As the results showed, none of the compounds showed any anti-inflammatory effect. We believe that the sesquiterpenes are not responsible for anti-inflammatory activity. A key question is whether the fact that the proportion of compounds 1–10 in the ethyl acetate phase was very small had an effect, and whether other potentially active substances were in the remaining material. The second possibility is the non-specificity of the bioassays used. We speculate that perhaps this method is not suitable for this group of compounds. Next, we will use at least one alternative method (an anti-inflammatory effect through the inhibition of the activity of NF-κB by blocking the nuclear translocation of p65) to determine anti-inflammatory activity. Following this, we will be able to state with confidence that it is not sesquiterpenes that are responsible for the anti-inflammatory activity, and that other substances should be sought in this fraction (maybe the triterpene fraction) for their anti-inflammatory effect.

## 4. Materials and Methods

### 4.1. General Information

ESIMS was performed using an Agilent 1100 series LC/MSD ion trap mass spectrometer (Agilent, Santa Clara, CA, USA). The 1D- and 2D-NMR spectra were obtained in CD_3_OD and DMSO with TMS as the internal standard on Varian 500 MHz and Bruker AV500-III spetrometers (Bruker Corporation, Billerica, MA, USA). Column chromatography was performed over silica gel (160–200 mesh, Qingdao Marin Chemical, Inc., Qingdao, China), RP-18 reverse phase silica gel (43–60 μm), cyanopropyl silica gel (43–60 μm) and Sephadex LH-20 (Pharmacia Biotech AB, Uppsala, Sweden). LPLC experiments were performed with Combiflash with a UV-detector (ISCO Companion, Lincoln, NE, USA). HPLC experiments were performed using a Waters 2545 system with 2998 diode array detector (Waters Corporation, Milford, MA, USA), using the Waters Sunfire and X-Bridge (250 mm × 10 mm i.d.) preparative columns packed with C18 (5 μm) (Alltech Associates, Inc., Bannockburn, IL, USA). TLC was carried out with glass precoated silica gel GF254 glass plates. Spots were visualized under UV light and by spraying with 8% H_2_SO_4_ in 95% EtOH followed by heating.

### 4.2. Experimental Material

The whole herb of *Ainsliaea yunnanensis* was collected from Chuxiong city, Yunan province, China. It was identified by Lu Jin Mei, Associate Researcher at the Kunming Institute of Botany, Chinese Academy of Sciences, Yunnan, China. The specimen had been deposited at Beijing Union University, and the Beijing Key Laboratory of Bioactive Substances and Functional Foods, Beijing, China.

### 4.3. Extraction and Isolation

The dried whole herb of *Ainsliaea yunnanensis* (10.0 kg) was ground into powder and extracted with aqueous EtOH (95%, 80%, and 70% sequentially) for 120 min under sonication. The total extraction of aqueous EtOH was suspended in water and then partitioned with petroleum ether, EtOAc and normal butanol respectively. The EtOAc portion (123.3 g) and normal butanol portion (420 g) were fractionated via silica gel column chromatography, eluting with a gradient of CHCl_3_-MeOH (30:1, 20:1, 10:1, 8:1, 6:1, 5:1, 4:1, 2:1, 0:100) with 1/1000 acetic acid to give seven fractions, Y1–Y7.

The Y1 fraction was chromatographed over a silica gel gradient eluting with petroleum ether–acetone (50:1–2:1) to give fifteen fractions Y1-1–Y1-15. The Y1-9 was subjected to column chromatography over Sephadex LH-20 gel eluting with CHCl_3_-CH_3_OH (2:1) to six fractions Y1-9-1–Y1-9-6. The Y1-9-2 fraction was eluted using a stepwise gradient of CHCl_3_-MeOH (100:0, 50:1, 40:1, 20:1), to afford nine fractions Y1-9-2-1–Y1-9-2-9. Next, the Y1-9-2-6 fraction was isolated with a gradient of petroleum ether–acetone (10:1–2:1) by LPLC using normal phase cyanopropyl silica, to give five fractions Y1-9-2-6-1–Y1-9-2-6-5. Finally, the Y1-9-2-6-4 fraction was purified by preparative HPLC using Waters sunfire reversed-phase C18 chromatographic column, eluting with MeOH-H_2_O (35:65, 18.0 mL/min), with a UV detection wavelength of 206 nm to collect peaks at 12.4 min, 16.7 min and 17.7 min to yield compound **5** (27.9 mg), compound **6** (27.8 mg) and compound **7** (30.6 mg), respectively.

The Y2 fraction was also chromatographed over silica gel gradient eluting with CHCl_3_-MeOH (100:1–10:1) to give five fractions Y2-1–Y2-5. The Y2-1 fraction and Y2-2 fraction were subjected to column chromatography over Sephadex LH-20 gel eluting with CHCl_3_-CH_3_OH (2:1) to give five fractions Y2-1-1–Y2-1-5 and four fractions Y2-2-1–Y2-2-4, respectively. The Y2-1-4 fraction was isolated with a gradient of petroleum ether–acetone (20:1–5:1) by LPLC using normal phase silica gel, to give five fractions Y2-1-4-1–Y2-1-4-5. The Y2-1-4-3 fraction was heated and dissolved in MeOH. Upon cooling the white precipitate was filtered to afford compound **1** (8.0 mg). The Y2-2-2 fraction was eluted using a stepwise gradient of CHCl_3_-MeOH (20:1–5:1), to afford six fractions Y2-2-2-1–Y2-2-2-6. Next, the Y2-2-2-5 fraction was purified by preparative HPLC using Waters sunfire reversed-phase C18 chromatographic column, eluting with MeOH-H_2_O (55:45, 18.0 mL/min), with a detection wavelength of 206 nm to collect peaks at 11.3 min, 12.0 min and 13.2 min to yield compound **4** (7.9 mg), compound **9** (52.0 mg) and compound **10** (39.6 mg), respectively, and the Y2-2-2-6 fraction was purified by preparative HPLC using Waters sunfire reversed-phase C18 chromatographic column, eluting with MeOH-H_2_O (57:43, 18.0 mL/min), and 206 nm detection wavelength to collect peak at 7.2 min to yield compound **8** (77.9 mg).

The Y3 fraction was subjected to column chromatography over Sephadex LH-20 gel eluting with CHCl_3_-CH_3_OH (2:1) to give twelve fractions Y3-1–Y3-12. The Y3-5 fraction was isolated with a gradient of petroleum ether–acetone (20:1–4:1) by LPLC using normal phase silica gel, to give six fractions Y3-5-1–Y3-5-6. Finally, the Y3-5-3 fraction was purified by preparative HPLC using Waters X-bridge reversed-phase C18 chromatographic column, eluting with MeOH-H_2_O (54:46, 18.0 mL/min), and 210 nm detection wavelength to collect peaks at 11.0 min and 14.0 min respectively to yield compound **3** (5.0 mg) and compound **2** (9.6 mg).

Ainsliatone A acid (**1**): white amorphous powder; [α]^25^_D_-295 (c 0.004, CH_3_OH); IR Vmax 3579, 3311, 2958, 2931, 1703 cm^−1^; ^1^H-NMR (MeOD, 500 MHz) and ^13^C-NMR data (MeOD, 125 MHz), see Table 1. HR-ESI-MS *m*/*z* 269.1382 [M + H]^+^ (calcd. for C_14_H_21_O_5_^+^, 269.1382).

Alatoside M (**2**): white amorphous powder; [α]^25^_D_-115 (c 0.003, CH_3_OH); IR Vmax 3440, 3218, 2933, 2917, 1693 cm^−1^; ^1^H-NMR (MeOD, 500 MHz) and ^13^C-NMR data (MeOD, 125 MHz), see Table 1. HR-ESI-MS *m*/*z* 435.1985 [M + Na]^+^ (calcd. for C_21_H_31_O_8_Na^+^, 435.1989).

Alatoside N (**3**): white amorphous powder; [α]^25^_D_-10 (c 0.003, CH_3_OH); IR Vmax 3422, 2924, 1710, 1383 cm^−1^; ^1^H-NMR (MeOD, 500 MHz) and ^13^C-NMR data (MeOD, 125 MHz), see Table 1. HR-ESI-MS *m*/*z* 379.2090 [M + Na]^+^ (calcd. for C_19_H_32_O_6_Na^+^, 379.2091).

4β,15-dihydrozaluzanin C (**4**): white amorphous powder; [α]^25^_D_10 (c 0.003, CH_3_OH); IR Vmax 3397, 2918, 2901, 1760 cm^−1^; ^1^H-NMR (DMSO, 500 MHz) and ^13^C-NMR data (DMSO, 125 MHz), see Table 1. HR-ESI-MS *m*/*z* 433.1832 [M + Na]^+^ (calcd. for C_21_H_29_O_8_Na^+^, 433.1833).

### 4.4. Anti-Inflammatory Activity Assays

Bone marrow-derived macrophage (BMDM) cell lines were obtained from ATCC (Rockefeller, MD, USA). The cell line was seeded in 96-well microtiter plates and maintained in a DMEM high sugar medium for 12 h after they were digested with EDTA and Trypsin. Then LPS (50 ng/mL, Invivogen) was added and stimulated for 4 h. The supernatant was discarded, and the compounds **1**–**10** (10 μmol/L) were added. After one hour, the nigericin (Sigma, Shanghai, China) was added and stimulated for 45 min and the samples were collected.

The supernatant of the cell culture was centrifuged at 3500 r/min and kept in a refrigerator overnight after the addition of TCA. The next day, they were centrifuged at 13000 r/min and 4 °C for 15 min. The supernatant was discarded, and the cells were washed once with ice-cold acetone. Next, 1× loading buffer 40 μL was added after acetone vaporizing. Finally, they were shaken, boiled and cooled to obtain the supernatant samples, which detected caspase-1 activity by using the Caspase 1 Activity Assay Kit and Promega GloMax 20/20 Luminescence detector.

## 5. Conclusions

Four new sesquiterpenoids, ainsliatone A acid (**1**), alatoside M (**2**), alatoside N (**3**), 4β,15-dihy drozaluzanin C (**4**) and six known abundant sesquiterpenoids, ainsliatone A (**5**), 8α-hydroxy-11α,13-hydroxyzaluanin C (**6**), 3,8-hydroxy-10(14)-guaien-6,12-olide (**7**), ixerin W (**8**), glucozaluza nin C (**9**), picriside B (**10**), were isolated from the EtOAc phase of the ethanolic extract of *Ainsliaea yunnanensis*. All compounds were tested for their anti-inflammatory effect in in vitro experiments through the inhibition of the activity of the NLRP3 inflammasome by blocking the self-slicing of pro-caspase-1, which was induced by nigericin, and then the secretion of mature IL-1β, mediated by caspase-1, was suppressed. As the results showed, none of the compounds showed any anti-inflammatory effect.

## Figures and Tables

**Figure 1 molecules-24-01701-f001:**
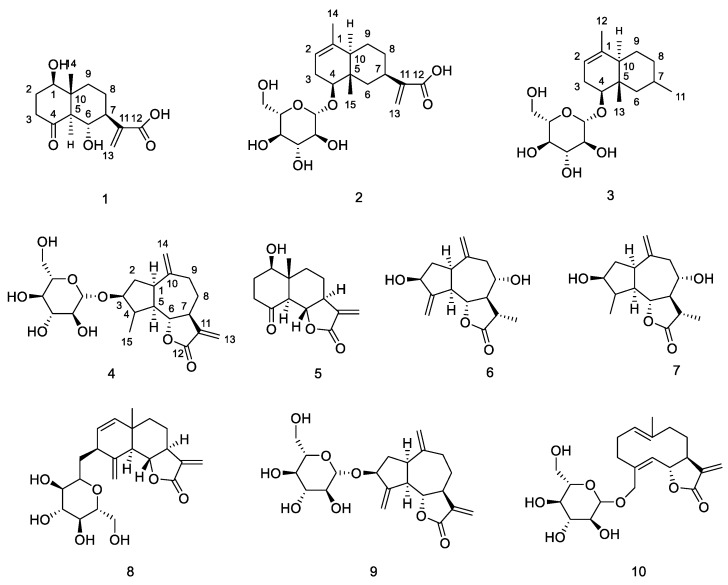
Chemical structures of Compounds **1**–**10**.

**Figure 2 molecules-24-01701-f002:**
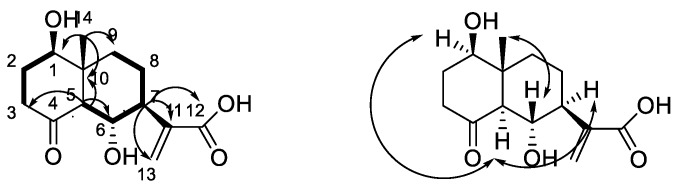
Main ^1^H-^1^H COSY (bold lines), HMBC (arrows) and NOE (double arrows) correlations of compound **1**.

**Figure 3 molecules-24-01701-f003:**
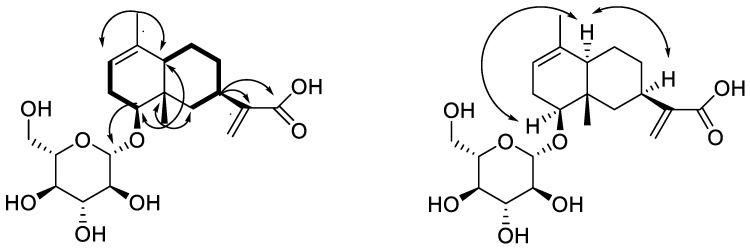
Main ^1^H-^1^H COSY (bold lines), HMBC (arrows) and NOE (double arrows) correlations of compound **2**.

**Figure 4 molecules-24-01701-f004:**
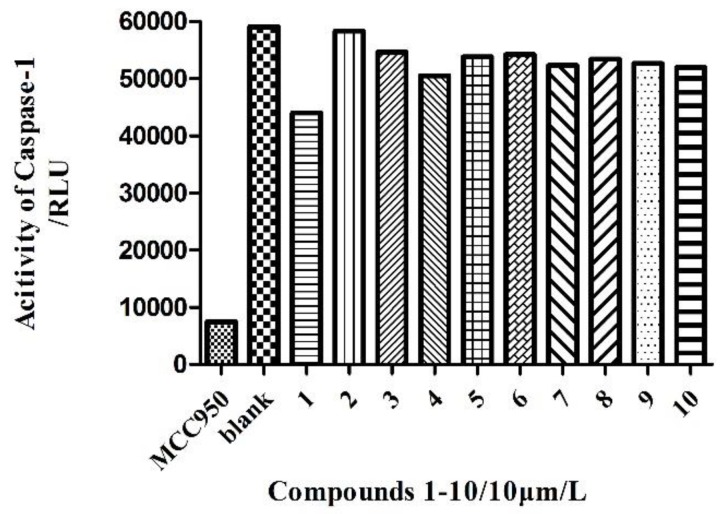
LPS-primed iBMDMs treated with 10 μm/L of compounds **1**–**10**, and then stimulated with nigericin. The activity of caspase-1 was analyzed in the supernatant of BMDMs by the Caspase-Glo^®^ 1 Inflammasome Assay. RLU, recombinant luciferase, which is proportional to caspase-1 activity.

**Table 1 molecules-24-01701-t001:** ^1^H and ^13^C-NMR spectral data for compounds **1**–**4**.

	1		2		3		4	
Pos.	δ_H_	δ_C_	δ_H_	δ_C_	δ_H_	δ_C_	δ_H_	δ_C_
**1**	3.80, dd, (10.0, 5.0)	77.3	-	136.2	-	136.3	2.76, m	42.1
**2**	1.84, m 2.08, m	31.6	5.31, br s	120.6	5.30, br s	120.6	1.69, dd, (12, 10) 2.14, dd, (13.5, 7.0)	37.3
**3**	2.21, m 2.53, m	40.6	2.03, m 2.41, m	29.9	2.00, m 2.40, m	29.0	3.56, dd, (8.0, 15.5)	86.2
**4**	-	212.6	3.72, dd, (10.0. 6.5)	82.2	3.67, dd, (10.0, 6.5)	82.2	1.85, m	44.1
**5**	1.64, m	62.2	-	37.9	-	37.9	1.91, m	49.5
**6**	4,14, t, (10.0)	67.6	1.26, m 2.40, m	36.5	1.25, m 2.37, m	36.3	3.93, t, (10)	86.2
**7**	2.41, m	48.4	2.02, m	41.3	1.95, m	42.5	2.78, m	46.8
**8**	1.64, m	28.1	1.50, m 1.67, m	28.1	1.57, m 1.72, m	25.3	1.24, m; 2.25, m	30.6
**9**	1.36, m 1.81, m	37.5	1.26, m 1.84, m	30.2	1.25, m 1.72, m	29.8	1.97, m; 2.54, m	35.7
**10**	-	45.7	2.04, m	37.9	2.02, m	48.3	-	149.5
**11**	5.67, s 6.23, s	125.9	-	147.9	1.13, d, (7.0)	14.7	-	140.1
**12**	-	170.5		170.9	1.58, s	21.0	-	169.6
**13**	-	144.1	5.58, s 6.14, s	123.0	0.81, s	10.7	5.59, d, (2) 6.00, d, (2.5)	119.5
**14**	0.76, s	12.2	1.58	21.0	-	-	4.95, d, (5) 4.99, d, (5)	112.7
**15**	-	-	0.85	10.9	-	-	1.15, d, (10)	18.1
**Glu-1**	-	-	4.30, d, (8.0)	101.5	4.30, d, (7.5)	101.5	4.19, d, (7.5)	104.1
**Glu-2**	-	-	3.14, m	75.2	3.14, m	75.2	2.95, m	73.6
**Glu-3**	-	-	3.30, m	78.2	3.31, m	78.2	3.16, m	76.8
**Glu-4**	-	-	3.27, m	71.9	3.27, m	71.9	3.04, m	70.2
**Glu-5**	-	-	3.25, m	77.8	3.22, m	77.8	3.08, m	76.8
**Glu-6**	-	-	3.65, dd, (12.0, 6.5) 3.85, dd, (11.5, 2.0)	63.0	3.67, dd, (12.0, 6.5) 3.85, dd, (11.5, 2.0)	63.0	3.42, dd, (11.5, 6.0) 3.65, d, (9.5, 4.5)	61.2

^1^H and ^13^C data were measured in CD_3_OD for compound **1**–**3** and in DMSO for compound **4** at 500 and 125 MHz. The assignments were based on the ^1^H-^1^H COSY, HMQC, and HMBC experiments.

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
