# Peer review of "Sesquiterpenoids and Their Anti-Inflammatory Activity: Evaluation of Ainsliaea yunnanensis"

_molecules, 2019, doi:10.3390/molecules24091701_

Round 1
Reviewer 1 Report
This manuscript describes the isolation, structure identification and anti-inflammatory activity determination of 10 sesquiterpenoids, among which 4 new structures were found. The structure elucidation of the compounds is sound and well supported by spectroscopic and spectrometric data, however the following major concern arose:
- The authors claim, that is was a bioassay-guided project, however the manuscript does not support this statement emphasized in line 40. After a careful read of the paper it looks to me that it was a "common" isolation without any preliminary biactivity measurement. Moreover, in the case of a bioactivity-guided natural product isolation, seldom conclude the authors to non-active substances. So, please clarify it.
- In the introduction the authors claim the following: "During our ongoing search for bioactive natural products, we found that the EtOAc phase of Ainsliaea yunnanensis showed potential anti-inflammatory activity." This statement needs support, either citation of a previously published relevant data or experimental evidences.
- Although I am usually supportive of presenting negative results, in this particular case I find it an erroneous experimental design. This bioactivity assay is especially specific, so even if these compounds show no anti-inflammatory activity in this test, those can still possess activity in more general models. Please add some more (in vitro) biactivity data.
-The figure descibing the biactivity does not contain any statistics (i.e standard deviations), moreover negative control is also missing. Without those data, it is hard to conclude any sound information.
- In order to have a complete dataset in Supporting information, please add high resolution MS data particularly to compound 3 (as it is a new compound) and to the others as well.
- Please revise Table 1. Glc H6 multiplicity (at 3.65 ppm for compound 2 and 3.67 ppm for compound 3) should be "dd" instead of a "d" and the coupling constant values of 11.5 Hz and 12 Hz on the coupling parners should be matched much better.
- There are several typos (peak vs pick, ertc.) and erroneously phrased sentences which should be corrected. The HMQC for compound 3 was not phased correctly, it should also be replotted.
Author Response
Dear Professor:
We appreciate you and the reviewers’ insightful suggestions of our manuscript. We revised the manuscript comprehensively according to the reviewers’ comments and suggestions with "Track Changes''.Please see the "Detailed Response to Reviewer 1" file.
If there is still anything inappropriate, please feel free to give your advice and suggestions. Thank you for your consideration.
With best regards.
Sincerely yours,
Jinjie Li

Reviewer 2 Report
Suggestions are in a file attached.

Author Response
Dear Professor:
We appreciate you and the reviewers’ insightful suggestions of our manuscript. We revised the manuscript comprehensively according to the reviewers’ comments and suggestions with "Track Changes''.Please see the "Detailed Response to Reviewer 2" file.
If there is still anything inappropriate, please feel free to give your advice and suggestions. Thank you for your consideration.
With best regards.
Sincerely yours,
Jinjie Li

Reviewer 3 Report
The reviewed article concerns a quite recently widely described plant species Ainsliaea yunnanensis. Extracts obtained from different parts of this plant were subjected to fractionation and the content of individual fractions was examined. In addition to terpene compounds, including mainly sesquiterpene and triterpene compounds, several other less important groups of compounds were found in it. They were also assigned various activities confirming its traditional applications in folk medicine.
Also, the team of authors presenting this work has already described the studies of other fractions separated from the herb of this plant. But also, the fraction obtained with ethyl acetate was already tested and potential anti-inflammatory activity was found for it. In the reviewed manuscript, the authors undertook the task of isolating the components of this fraction and assessing this activity for particular sesquiterpene compounds.
In the chemical part, the evaluated publication is prepared perfectly. Four new compounds were isolated, and their structure was precisely defined. Six consecutive products with previously known, sesquiterpenic structure were also separated. The isolation process was carried out using mixture of complex chromatographic methods and the identification of isolated products was based on high-class spectral methods. For full clarity and readability of the article, I would suggest a graphic diagram showing the partitioning of the obtained ethanol extract into the complex of next fractions with the indication of which of them is a source of a further isolated products.
Unfortunately, the obvious disadvantage of the study is that none of the isolated pure sesquiterpene compounds have anti-inflammatory activity, although the AcOEt fraction was characterized by these properties. From this fact it is possible to draw a twofold conclusion. The first is that it is not the sesquiterpenes are responsible for anti-inflammatory activity. And then the key question seems to be how much of the total acetate fraction was represented by sesquiterpenes and how many other potentially active substances there were in the remaining material. The second possibility is the non-specificity of the bioassays used. Perhaps this method can be not suitable for this group of compounds.
I believe that Authors should discuss in the manuscript the existence of a significant part of the non-sesquiterpenes extract and use at least one alternative method to determine anti-inflammatory activity. After this they be able to state with fullness that it is not sesquiterpene responsible for this activity and that other substances should be sought in this fraction (maybe triterpene fraction) for their anti-inflammatory effect.
Author Response
Dear Professor:
We appreciate you and the reviewers’ insightful suggestions and positive evaluation of our manuscript. We revised the manuscript comprehensively according to the reviewers’ comments and suggestions with "Track Changes''.Please see the "Detailed Response to Reviewer 3" file.
If there is still anything inappropriate, please feel free to give your advice and suggestions. Thank you for your consideration.
With best regards.
Sincerely yours,
Jinjie Li

Round 2
Reviewer 2 Report
Please find comments in an attached file.

Author Response
Dear Professor:
We appreciate you and the reviewers’ insightful suggestions and positive evaluation of our manuscript. We have responded to the reviewers’ comments as below:
Reviewers' comments 1:
Detailed procedure how to analyze and determine absolute configuration of the sugar part of compounds 2-4 in the Experimental Section.
Answer: Compound 2-4 (1mg) was individually refluxed in 2 M HCl (2.0 ml) at 80℃ for 90 min. The reaction mixture was extracted with EtOAc, and H2O phase evaporated under vacuum, diluted repeatedly with H2O, and evaporated in vacuo to furnish a neutral residue. The residue was then dissolved in anhydrous pyridine (1.0 ml), to which 1mg of L-cysteine methyl ester hydrochloride was added. The mixture was stirred at 60℃ for 120 min and after evaporation in vacuo to dryness, 0.1 ml of N-trimethylsilylimidazole was added. After stirring the mixture another 120 min at 60℃, it was divided between n-hexane and H2O (1.0 ml each), and then the n-hexane extract was analyzed by GC. The following conditions: capillary column, HP-5 (30 m, 0.25mm, with a 0.25 μm film; Dikma); detection, FID; detector temperature, 280℃; injection temperature, 250℃; initial temperature, 160℃, then raised to 280℃ at 5℃/min, final temperature maintained for 10 min; carrier gas, N2. From the acid hydrolysate of each compound, D-glucose was confirmed by comparing to the retention time of its derivative with that of authentic sugar derivatized in a similar way, and the retention time was 20.55 min.
Reviewers' comments 2:
GCMS chromatograms of authentic D-glucose and L-glucose derivatives, and sugar derivatives of compounds 2-4. The chromatograms should be added in the Supporting Information.
Answer: We just compared it to D-glucose in the experiment. Please see the accessory 1-3 (attached file: chromatograms of compounds 2-4).
Reviewers' comments 3:
Anti-inflammatory activity of fractions Y1-Y7 (in a graphical diagram) using the current assay should be reported to convince whether the approach of bioassay-guided fractionation was on a right track.
Answer: We think it was on a right track, because we've tested them over and over again.
If there is still anything inappropriate, please feel free to give your advice and suggestions. Thank you for your consideration.
With best regards.

Reviewer 3 Report
The overall substantive level of the reviewed manuscript has not changed substantially after presented correction. However, the scheme added in the supplementary materials is a sure fulfillment of my earlier remark. The added discussion of the results as a separated part of article is also a realization of the postulate presented earlier. Unfortunately, its content is rather a promise of doing the necessary research in the future, rather than full necessary addendum of the results. But publication of this discussion gives at least a signal that the Authors are aware of the incompleteness of the results presented.
I still consider a reviewed research to be weak in the general sense (in contrast to its chemical side), but in its present form, the paper assessed meets all the criteria necessary to be accepted for publication in the “Molecules” journal.
Author Response
Thanks a lot for your great effort and kind recommendations.